# Fast, In Vivo Model for Drug-Response Prediction in Patients with B-Cell Precursor Acute Lymphoblastic Leukemia

**DOI:** 10.3390/cancers12071883

**Published:** 2020-07-13

**Authors:** Anton Gauert, Nadine Olk, Helia Pimentel-Gutiérrez, Kathy Astrahantseff, Lasse D. Jensen, Yihai Cao, Angelika Eggert, Cornelia Eckert, Anja I.H. Hagemann

**Affiliations:** 1Department of Hematology/Oncology, Charité–Universitätsmedizin Berlin, 13353 Berlin, Germany; anton.gauert@charite.de (A.G.); nadine.olk@charite.de (N.O.); heli.pimentel@gmail.com (H.P.-G.); kathy.astrahantseff@charite.de (K.A.); angelika.eggert@charite.de (A.E.); cornelia.eckert@charite.de (C.E.); 2German Cancer Consortium (DKTK)—German Cancer Research Center (DKFZ), 69120 Heidelberg, Germany; 3Department of Health, Medical and Caring Sciences, Linköping University, 58183 Linköping, Sweden; lasse.jensen@liu.se; 4Department of Microbiology, Tumor and Cell Biology, Karolinska Institutet, 17165 Stockholm, Sweden; yihai.cao@ki.se

**Keywords:** pediatric cancer, personalized therapy, BCP-ALL, zebrafish, patient-derived xenograft

## Abstract

Only half of patients with relapsed B-cell precursor (BCP) acute lymphoblastic leukemia (ALL) currently survive with standard treatment protocols. Predicting individual patient responses to defined drugs prior to application would help therapy stratification and could improve survival. With the purpose to aid personalized targeted treatment approaches, we developed a human–zebrafish xenograft (ALL-ZeFiX) assay to predict drug response in a patient in 5 days. Leukemia blast cells were pericardially engrafted into transiently immunosuppressed *Danio rerio* embryos, and engrafted embryos treated for the test case, venetoclax, before single-cell dissolution for quantitative whole blast cell analysis. Bone marrow blasts from patients with newly diagnosed or relapsed BCP-ALL were successfully expanded in 60% of transplants in immunosuppressed zebrafish embryos. The response of BCP-ALL cell lines to venetoclax in ALL-ZeFiX assays mirrored responses in 2D cultures. Venetoclax produced varied responses in patient-derived BCP-ALL grafts, including two results mirroring treatment responses in two refractory BCP-ALL patients treated with venetoclax. Here we demonstrate proof-of-concept for our 5-day ALL-ZeFiX assay with primary patient blasts and the test case, venetoclax, which after expanded testing for further targeted drugs could support personalized treatment decisions within the clinical time window for decision-making.

## 1. Introduction

While risk-stratified therapy optimization trials have pushed event-free survival to more than 80% in newly diagnosed B-cell precursor (BCP) acute lymphoblastic leukemia (ALL) in the last decades, only 50% of children and young adults who experience relapse survive disease, and side effects of standard treatment with cytostatic drugs are tremendous [1]. Targeted immunotherapy has become an attractive treatment option for leukemias to enhance leukemic cell eradication and reduce polychemotherapy-associated toxicity. Response rates and outcomes were significantly improved in patients with relapsed/refractory BCP ALL by treatment with the CD22-directed toxin-conjugated monoclonal antibody, inotuzumab-ozogamicin or the CD19-directed, T cell-engaging bispecific antibody, blinatumomab [2,3]. Although, resistance can also develop to targeted immunotherapies via target loss (CD19/CD22) on leukemic cells, or the development of mechanisms preventing induction of apoptosis in the case of inotuzumab-ozogamicin. In vitro assays assessing proliferation of leukemia cells from bone marrow biopsies have improved over the years [4], and have entered clinical use for severe cases within trials for relapse/refractory disease [5]. Nevertheless, Schramm et al. recently demonstrated a lack of correlation between in vitro and in vivo drug response, as well as a lower predictive value of in vitro drug testing, reflecting an intrinsic limitation of this methodology that prevents its use for treatment stratification in future trials. Here, a simple fast in vitro model was prospectively applied for response prediction and treatment stratification accordingly using a 2D monoculture of patient samples and the methyl-thiazoltetrazolium viability assay [6]. The most widely used preclinical in vivo models for hematological malignancies are human tumor xenografts in immunodeficient mice [7]. Between 10 and 100 cells from patients with ALL are sufficient to reconstitute leukemia in immunodeficient NOD/scidIL2Rgnull mice after intrafemoral injections with up to 100% engraftment [8,9]. Patient-derived xenografts (PDX) of leukemic blasts in mice have been used and improved to retrospectively screen for efficacy of new drugs or combination treatments in relation to the genetic background of leukemia types since 1995 (reviewed in [10]). The genomic landscape of leukemias at the population level closely resembles that preserved in the mouse PDX models, as is the inherent heterogeneity in leukemic blasts derived from a single patient [8,9,11]. However, similar to their endemic human bone marrow niche, human leukemia blasts in the murine setting are exposed to both favorable and unfavorable conditions for engraftment that may exert distinct pressures for subclone selection [12]. These characteristics provide a strong rationale for performing preclinical drug screens on panels of existing PDX models to preclinically investigate efficacy in a sample population reminiscent of intra- and inter-patient heterogeneity. While this approach presents data generalized for the response to a group of patients with a specific leukemia type, it fails to deliver personalized response information for single patients. Successful engraftment of human ALL cells takes several weeks to several months, too long for clinical applications in real time [13]. Low cost, space efficiency, external embryonic development and rapid growth make zebrafish an excellent alternative as a transplant host organism to speed in vivo testing. In recent years, the zebrafish embryo and adult received increasing attention as an avatar for human cancer cells, but mostly originating from solid tumors. Many cell lines derived from human cancers, including breast cancer [14], melanoma [15] or gastric cancer [16] among others, have successfully been transplanted into zebrafish embryos (reviewed in [17]). The first successful PDX performed in 2-day old zebrafish embryos engrafted gastrointestinal human tumors [18], and were followed by xenografting patient-derived samples of neuroendocrine tumors [19] and colorectal cancers [20]. Bentley et al. were the first to use zebrafish embryos as hosts for two patient-derived T-cell ALL samples that were transplanted into the yolk sac of 2-day old embryos to determined quantitative drug response to rhapamycin and compound E, which was assessed by microscopically counting fluorescently labeled cells in dissociated host/graft cell suspensions [21]. They were followed by only one study to date, xenografting primary human leukemia cells (acute myeloid leukemia, AML) from four patients [22] to test drug response for the phytotoxin, stemphol. To date, only two other studies have evaluated graft cell responses in light of the corresponding clinical course in the patients. Fior et al., who reported a correlation in four of five xenografted colorectal cancer samples [20]), and Wu et al., who reported that the gastric cancer sample from one patient correlated with the clinical course [16]. Predicting individual responses in patients with BCP-ALL to defined drugs would greatly help therapy stratification to balance the efficient eradication of cancer cells while minimizing unnecessary side effects.

The US Food and Drug Administration approved the BCL2 apoptosis regulator inhibitor, venetoclax, to treat adult patients with chronic lymphoblastic leukemias and acute myeloid leukemia. Venetoclax also recently entered a phase I trial (EudraCT 2017-000439-14, NCT03236857) as either single-agent therapy or in combination with chemotherapy for relapsed/refractory cancers (including BCP-ALL) in pediatric and young adult patients [23]. With the purpose to improve individual targeted treatment approaches, we present the human–zebrafish xenograft (ALL-ZeFiX) assay as a proof-of-concept for in vivo drug response prediction for the test case, venetoclax, from patient-derived leukemic blasts in 5 days. 

## 2. Results

### 2.1. Engraftment Site Strongly Influences BCP-ALL Graft Viability

We established a protocol for xenotransplantation of primary human BCP-ALL cells, building on previous publications [21,24,25,26] with a number of essential improvements to enable survival and expansion of human BCP-ALL cells. Human BCP-ALL cells were implanted into zebrafish embryos, which then could be bathed in various drug dilutions for the ALL-ZeFiX assay. Zebrafish are able to adapt to temperatures between 18 °C and 38 °C in their natural habitat, while usual fresh water temperature in captivity is 27 °C. To establish assay conditions, we first tested temperatures that BCP-ALL cells in 2D cultures could withstand. Growth of engrafted human BCP-ALL cell lines (SEM, Nalm-6 and RCH-ACV) was comparable to growth in 2D culture at 35 °C or 37 °C (Figure 1A). We compared the yolk sac and pericardium as transplantation sites in wildtype zebrafish embryos at 2 days post-fertilization. Graft cells were labeled with a fluorescent proliferation marker (CellTrace Violet) prior to transplantation to support quantification of graft cell proliferation over time. CD19 expression was also assessed to assure BCP-ALL blast cell identity. After 3 days at 35 °C, 10–20 engrafted embryos were pooled and dissociated into single-cell suspensions, in which leukemia cell viability, proliferation and total human leukemia cell numbers were flow cytometrically analyzed (Figure 1D and Appendix A) to assess engraftment success and graft expansion. Viability and CD19 expression were directly measured from the CellTrace Violet-labeled graft cell population (see gating strategy, Appendix A). Counting only CD19-positive cells assured that we only assessed engrafted human leukoblast expansion. The number of cell divisions of the CD19-positive leukoblasts was calculated from the mean fluorescence intensity of the CellTrace Violet proliferation marker, which is reduced by half in successive daughter cell generations. In contrast to previous reports [21,24,25], cells transplanted into the yolk sac showed poor viability, while engrafting cells into the pericardium significantly improved viability (Figure 1B and Appendix A). Graft viability and proliferation rates with our optimized ALL-ZeFiX assay conditions were comparable to cells in conventional 2D cultures.

### 2.2. Graft Expansion Requires Transient Host Immunosuppression

Although 80% of graft cells were viable throughout the 3-day testing period, graft expansion was limited. Graft cells underwent 3 to 3.5 cell divisions in 3 days (Figure 1A), predicting 2400–4000 cells from the 300–500 cells that were engrafted. However, grafts averaged only 180–1100 after 3 days. To understand this discrepancy, we microscopically monitored Nalm-6 grafts labeled with the stable lipophilic carbocyanine fluorescent lineage tracer, DiO (Appendix A). After 3 days of engraftment, Nalm-6 cells had disseminated from the injection site and total graft cell numbers were diminished (Figure 1C, quantified in Appendix A). We reasoned that the zebrafish innate immune response might interfere with graft survival and growth [27]. To test this hypothesis, endogenous expression of Spi1 and Csf3r, two proteins involved respectively in macrophage and neutrophil differentiation, was transiently suppressed by injecting morpholino antisense oligonucleotides into host embryos at the one-cell stage [28,29,30]. We confirmed the transient immunosuppression window provided by dual-mopholino knockdown in our macrophage *Tg*(*mpeg1:mCherry*) and neutrophil *Tg*(*mpx:EGFP*) reporter lines. These zebrafish embryos were devoid of neutrophils and macrophages until 4 days after morpholino injection, as shown exemplarily for the *Tg*(*mpeg1:mCherry*) line in Appendix A. Cell tracing experiments 1 and 3 days after BCP-ALL cell engraftment into these immunosuppressed zebrafish embryos as well as graft cell numbers quantified flow cytometrically for three different cell lines demonstrated significantly better graft expansion with immunosuppression (Figure 1C, Appendix A). Experiments with single morphants suggest the necessity of the double morphants to reach sufficient graft cell survival, even though *spi1* knockdown had a more pronounced effect on graft cell survival than *csf3r* knockdown (Appendix A). Transplantation of Nalm-6 into zebrafish transgenic lines with fluorescently trackable endogenous macrophages and neutrophils also revealed clear attraction of macrophages to the transplantation site one day after injection (Appendix A). Approximately 38% of all macrophages present at the graft site, but only 15% of neutrophils, directly interacted with Nalm-6 cells at the graft site, as quantified from high-resolution 3D confocal images of six host embryos two days after injection (Appendix A). Our data confirm that morpholino-based transient immunosuppression is necessary for optimal graft survival and growth in the ALL-ZeFiX assay. Therefore, all further experiments using the ALL-ZeFiX assay were conducted in morpholino-based transiently immunosuppressed zebrafish embryos.

### 2.3. BCP-ALL Graft Response to Venetoclax Reflects 2D Culture Sensitivity

We next assessed treatment response to the small molecule BCL2 inhibitor, venetoclax, in our ALL-ZeFiX assay engrafted with the BCP-ALL cell lines, SEM and RCH-ACV. SEM cells in 2D cultures were highly responsive to venetoclax after 48 h, with an IC50 of <10 nM, whereas RCH-ACV cells responded poorly (IC50 ~ 1000 nM, Figure 2A and Appendix A). Newly engrafted zebrafish embryos were transferred to a 96-well plate (1 embryo/well) for venetoclax treatment (3 days). Venetoclax concentrations below 100 μM produced no obvious signs of toxicity in host embryos (Appendix A). Graft expansion and viability of engrafted SEM cells were assessed after 3 days in single-cell suspensions from 10–20 pooled embryos. BCP-ALL cells engrafted in zebrafish embryos were ~50-fold less sensitive to venetoclax than cells in conventional 2D culture (Figure 2B and Appendix A). This is not surprising in light of it being an in vivo assay requiring penetration across several cell layers in the embryo and graft. Drug response assessment in the ALL-ZeFiX assay was comparable to 2D cultures. 

### 2.4. BCP-ALL Blast Cells from Patients Can Be Expanded and Treated in Zebrafish Embryo Avatars

We used our validated ALL-ZeFiX assay (Figure 2C) to test graft cell expansion in a pilot group of 15 patient-derived BCP-ALL grafts (Appendix A). Engraftment of blast cells from 1 primary (patient 2) and 14 relapsed cases of BCP-ALL was successful in 9/15 ALL-ZeFiX assays (60%). Blast cell viability was 45–93% with 67–99% CD19-positive cells prior to engraftment, and 30–80% with 57–96% CD19-positive cells following 72-h engraftment in untreated controls (Figure 3 and Appendix A). Viable blast cells completed 1.5 to 3 divisions in the 3-day assay period (Appendix A). Blast cell viability was substantially better in the ALL-ZeFiX assay than in 2D cultures (viability ranged from 1–20%) in six of seven patient-derived samples. We treated seven patient-derived xenografts with three concentrations between 50 and 5000 nM (Figure 3) or 10 and 1000 nM (Appendix A) of venetoclax. In the ALL-ZeFiX assay, four of seven patient-derived BCP-ALL grafts responded very well to venetoclax (patients 1, 2, 3 and 6 with IC50 in the range of 50–500 nM, Figure 3,Appendix A), two responded well (patients 5 and 7 with IC50~5000 nM, Figure 3) and one responded poorly after an initial effect at 50 nM (patient 4, range of tested concentrations were not active enough to calculate an IC50, Figure 3). Engrafted cells from patient 5 harbored a *BCR-ABL* fusion and responded more strongly to the tyrosine kinase inhibitor, dasatinib (Appendix A). Engrafted cells from patient 4 responded more weakly to venetoclax in the ALL-ZeFiX assay with an initial response at 50 nM that was not enhanced with increasing doses (Figure 3), corresponding well with the poor patient response to venetoclax within an ongoing phase I trial (Appendix A). Engrafted cells from patient 7 responded to venetoclax (IC50 between 500 and 5000nM), in line with a good clinical response of this patient in the ongoing trial (Appendix A). To date, this patient is in remission more than one year after venetoclax treatment. The variability in response of the engrafted patient-derived blast cells reflects that our ALL-ZeFiX assay can capture the breadth of scope expected for preclinical testing of patient samples.

## 3. Discussion

In vitro assays assessing proliferation to preclinically assess drug efficacy on viability of leukemia cells from bone marrow biopsies have improved over the years [4], and have entered clinical use for severe single cases within trials for relapse/refractory disease [5]. One study has described co-culturing primary BCP-ALL blasts with mesenchymal stem cells to improve in vitro culture methods, but these demonstrated high variability of spontaneous apoptosis after 3 days in culture or strongly reduced proliferation [4]. Our own attempts to apply a method of 2D co-culture with patient derived mesenchymal stem cells have not yet been systematically compared to the ALL-ZeFiX model, however, we show here that viability of patient-derived BCP-ALL blasts is improved in our ALL-ZeFiX model in comparison to standard in vitro culture. Viability was so high, and with less variability in patient-derived samples in this small pilot cohort, that we speculate it could also improve on co-culture with mesenchymal stem cells, since the erratic apoptosis-inducing signals appear to be absent or at least reduced in the 72-h zebrafish model.

We instituted a number of improvements to xenografting protocols for human cancer cells in zebrafish. Using a method not previously applied for xenografting cancer cells, we improved BCP-ALL cell engraftment conditions by a morpholino-driven temporal immune suppression that blocked innate differentiation of both macrophage and neutrophils. While the use of a mutant zebrafish line lacking both macrophages and neutrophils would be preferable, none exist to our knowledge since knocking out both *spi1* and *csf3r* does not produce viable mutants (personal communication with Jean-Pierre Levraud). Fluorescently labeling graft cells with CellTrace Violet made assessing cell divisions in the 3-day treatment period possible, as the marker is inherited equally by daughter cells through several divisions. This methodology improves on commonly used proliferation assays assessing proliferative activity in a cell population only at a single time point [20,31]. Our data demonstrate that BCP-ALL cells find the pericardial cavity to be a more hospitable environment than the yolk sac. Interestingly, the group of Jason Berman, who demonstrated feasibility of T-cell ALL engraftment into the yolk sac of zebrafish embryos in 2015 [21], recently reported that the zebrafish embryonic yolk sac actually provides a hypoxic environment for graft cells [32]. The hypoxic conditions might explain why any BCP-ALL cells transplanted into the yolk sac survived. Our flow cytometric methods to assess total cell proliferation (CellTrace Violet) and viability (annexin V/7AAD) more accurately and independently quantify proliferation and viability in engrafted cells than the almost ubiquitously used microscopy-based methods applied to zebrafish xenograft models [20,31,33,34,35,36,37]. 

Xenograft models intended to support therapy decisions in personalized treatment approaches must ideally be feasible in one week from patient biopsy to support decisions for nonresponders to standard therapy. While engraftment of human leukemia cells is >70% successful in mice [8], engraftment and testing require 6 weeks to 6 months, making mice a less attractive option for clinical applications. In spite of the lower engraftment success of 60%, zebrafish embryos present a much more rapid option, even feasible for clinical applications in real-time. Clinical application of zebrafish avatars remains quite rare in the literature to date, with only two studies reporting treatment correlations with clinical records. Fior et al. presented feasibility for engrafting primary colorectal cancer cells into zebrafish embryos, then demonstrated that 4 of 5 patient-derived xenografts treated with the standard treatment protocol responded similarly to clinical records for the patients from whom the models were derived [20]. Wu et al. reported that engrafted gastric cancer from a single patient responded similarly to the standard treatment protocol as the patient [16]. We were interested if patient-derived BCP-ALL xenografts in zebrafish could be used to assist selection of promising targeted drugs in refractory or relapse cases, and began with the BCL2 inhibitor, venetoclax, for which treatment response in the ALL-ZeFiX assay correlated with responses in two patients with BCP-ALL receiving venetoclax treatment on our ward to date. Although engrafted leukemic blasts from patient 7 demonstrated low viability, the remaining viable cells did show a dose-dependent response to venetoclax with an IC50 10-fold higher than the IC50 for other patient-derived xenografts. These data are in line with the clinical course in this patient, who was diagnosed with a second BCP-ALL relapse refractory to both chemo- and immuno-therapy before being treated with venetoclax, and who remains in remission to date, one year after treatment. Treatment of engrafted leukemic blasts from patient 4 reflected the poor response during the treatment coarse. Zebrafish have received more attention during the last decade as an embryonic model host for xenografts, with nine publications exploring engraftment of other leukemic blast cell types to date [21,24,25,36,38,39,40,41]. The ALL-ZeFiX assay both provides proof-of-principle for BCP-ALL exgraftment and extends the clinical utility of drug testing in immunosuppressed zebrafish avatars with quantifiable endpoint analysis and a 5-day assay format.

## 4. Materials and Methods

### 4.1. Patients and Patient Samples

Approval for the use of patient samples in research for preclinical drug testing was provided within add-on studies to the ALL-REZ BFM 2002 trial (NCT00114348; 222/2001; December 2001) and the ALL-REZ BFM registry and biobank (EA2/055/12; July 2012) by the local medical research ethics committees, and to the IntReALL SR 2010 international trial (NCT01802814; EudraCT-Number: 2012-000793-30; July 2013) by the national authority (Landesamt für Gesundheit und Soziales). Informed consent was obtained from patients and/or guardians from the trial/registry in which they were enrolled. Patient 4 is enrolled in the phase I trial (trial registration: EudraCT 2017-000439-14; NCT03236857) testing venetoclax alone or in combination with chemotherapy in children and adolescent patients with relapsed/refractory cancers. Following the bone marrow biopsy (source of the cells tested in ALL-ZeFix) to diagnose the second relapse, patient 4 had received one course of venetoclax monotherapy followed by venetoclax in combination with vincristine and dexamethasone. The analyses presented here (Appendix A) were completed independently of the trial. Patient 7 was not enrolled in NCT03236857, but was treated according to recommendations in NCT03236857, and received two combined treatment courses according to protocol IIIb (cyclophosphamide, 6-thioguanine und cytarabine cycles in combination with venetoclax, Appendix A). Patient consent for sample use from patients 4 and 7 was obtained within the ALL-REZ BFM registry and biobank. Samples from patients 4 and 7 were collected at diagnosis of second relapse. The second relapse was diagnosed in patient 4 approximately one year after completion of first relapse treatment. The second relapse was diagnosed in patient 7 during ongoing disease progression that followed different courses of chemotherapy to treat the first relapse. Mononuclear cells were isolated using Ficoll density gradient centrifugation and viable frozen cells stored in liquid nitrogen in the biobank of the *ALL-REZ BFM* trials.

### 4.2. Zebrafish

Zebrafish were kept at 27 °C with a 14/10-h light/dark cycle. The Tüpfel Long Fin (TÜLF) wild-type line was used for all transplantations if not stated otherwise. Transgenic zebrafish *Tg*(*mpx:EGFP*)*^i114^* neutrophil reporter line was generated by Phil Ingham (Sheffield, UK) and *Tg*(*mpeg1:mCherry*) macrophage reporter line was generated by Graham Lieschke (Clayton, Australia). This project was approved by the national ethics authority, Landesamt für Gesundheit und Soziales (license number G0306/15). All studies were performed with zebrafish embryos ≤ 5 days post fertilization and did not fall under the Protection of Animals Act.

### 4.3. Cell Culture

SEM, RCH-ACV and Nalm-6 cell lines were from Leibniz Institute DSMZ-German Collection of Microorganisms and Cell Cultures GmbH and were kept in Roswell Park Memorial Institute (RPMI) 1640 medium (Thermo Fischer Scientific, Waltham, MA, USA) supplemented with 10% fetal calf serum (FCS, Merck, Darmstadt, Germany) and 1% penicillin/streptomycin (Thermo Fischer Scientific).

### 4.4. Morpholino Injection

To create a transient 4-day immunosuppressed condition in zebrafish embryos, 50 µM each of morpholino antisense oligonucleotides directed against the ATG start site of *spi1* (GATATACTGATACTCCATTGGTGGT) and *csf3r* (GAAGCACAAGCGAGACGGATGCCAT, both from Gene Tools, LLC, USA; sequence from [28,29,30]) were microinjected into 1–2 cell stage embryos in a total volume of 1 nL. 

### 4.5. Xenotransplantation

For cell lines from culture, 10^7^ BCP-ALL cells, or 10^6^ mononuclear cells isolated from frozen patient material, were resuspended in phosphate-buffered saline (PBS) prior to transplantation and labeled with CellTrace Violet (Thermo Fischer Scientific) according to the manufacturer’s instructions. CellTrace Violet was incubated with cells for 5 min at room temperature in the dark. Reaction was stopped with pre-warmed growth medium (RPMI + 10% FCS and penicillin/streptomycin). Cell pellets are filtered through a 10µm pluriStrainer^®^ fine filter (PluriSelect GmbH, Leipzig, Germany), washed and resuspended in 25 mL PBS. An aliquot of labeled cells served as proliferation control, and was flow cytometrically analyzed directly or cultured in parallel to transplantation experiments.

Dechorionated embryos were anesthetized in 0.03% Tricaine (Sigma-Aldrich, St. Louis, MO, USA) and transferred to an agarose-coated dish. Glass pipettes with a 20 mm outer diameter (BioMedical Instruments, Zöllnitz, Germany) connected to an air-pressure injector (IM-400, Narishige, Tokyo, Japan) were used to inject 300–500 human BCP-ALL cells at a concentration of 2 × 10^5^/μL into each embryo 48 h after fertilization. Embryos were allowed to recover in E3 medium (5 mM NaCl, 0.17 mM KCl, 0.33 mM CaCl, 0.33 mM MgSO_4_, pH 7.4) supplemented with 1% penicillin/streptomycin for 1 h at 28 °C before manually sorting for transplantation success on a M165 FC stereomicroscope (Leica Microsystems, Wetzlar, Germany). Embryos harboring engrafted human BCP-ALL cells were transferred to fresh E3 medium and incubated at 35 °C for up to 3 days.

### 4.6. Drug Response Assessment

Venetoclax (Selleckchem, Housten, TX, USA) stock was prepared in 100% dimethylsulphoxide (DMSO, Carl Roth, Karlsruhe, Germany). Dilutions of venetoclax were prepared with 0.5% DMSO in medium (RPMI + 10% FCS and penicillin/streptomycin) for cell culture or E3 medium supplemented with 1% penicillin/streptomycin for zebrafish embryos. Cells in culture were treated with venetoclax dilutions or vehicle in 6-well plates for 48 h, 0.5 × 10^6^ BCP-ALL cells per well. Successfully engrafted zebrafish embryos were treated in 96-well plates (1 embryo/well) containing 150 µL of venetoclax dilution or vehicle as control. Embryos were maintained at 35 °C for up to 72 h in the presence of drug or vehicle, then 10–20 embryos were pooled and dissociated for analyses. 

### 4.7. Embryo Dissociation and Flow Cytometric Analysis

Pooled zebrafish embryos from each treatment group were killed on ice then transferred to calcium- and magnesium-free Hank’s balanced salt solution (Sigma-Aldrich). After an initial mechanical dissociation (trituration through a yellow micropipette tip), embryo fragments were pelleted at 1500 rpm before being resuspended in 0.01% papain (Sigma-Aldrich), 0.1% dispase II (Sigma-Aldrich), 0.01% deoxyribonuclease I (AppliChem GmbH, Darmstadt, Germany) and 12.4mM MgSO_4_ (Sigma-Aldrich) in calcium- and magnesium-free Hank’s balanced salt solution for 15 min at room temperature with trituration through a micropipette tip every 5 min. Dissociated cells were centrifugated through a 20 µm fine mesh filter then resuspended in 140 mM NaCl, 3.33 mM CaCl_2_ and 10 mM Hepes. Immunostaining for flow cytometry was performed in parallel on all cell samples with Alexa Fluor^®^ 488 anti-human CD19 antibody (BioLegend, San Diego, CA, USA) for graft cell identity, and 7-AAD viability staining solution (eBioscience™ from Thermo Fischer Scientific) and fluorescein isothiocyanate-conjugated annexin A5 (BioLegend) to assess viability. The identity of single cells was determined by the fluorescence intensity for the 3 markers on a Becton Dickinson LSRFortessa™ X-20 flow cytometer and results were analyzed with FlowJo software (Becton Dickinson, Heidelberg, Germany).

### 4.8. Imaging

Life-engrafted zebrafish embryos were photographed with a Leica DFC 7000 T camera mounted on a Leica M165 FC stereomicroscope, and images were acquired using the Leica Application Suite X software. High-resolution 3D images of life embryos were acquired from a Nikon Spinning Disk Confocal CSU-X microscope. Immune and graft cell interactions were counted manually in 3D images stacked together with Fiji software [42]. 

## 5. Conclusions

Here we present the development of the ALL-ZeFiX assay and proof-of-concept evaluation for venetoclax treatment in patient-derived engrafted BCP-ALL cells. The ALL-ZeFiX assay provides individual response prediction in less than a week, with 60% engraftment success in fish embryos. We developed associated protocols to enable individual quantification of both cell divisions and viability in the graft to support reproducible response estimates. Xenograft responses to venetoclax were varied, as would be expected from patients with different clinical courses, and were in line with clinical courses in two patients. After expanded testing in further candidate drugs and an expanded cohort of patients with BCP-ALL, the ALL-ZeFiX assay has the potential to support treatment decisions in real time.

## Figures and Tables

**Figure 1 cancers-12-01883-f001:**
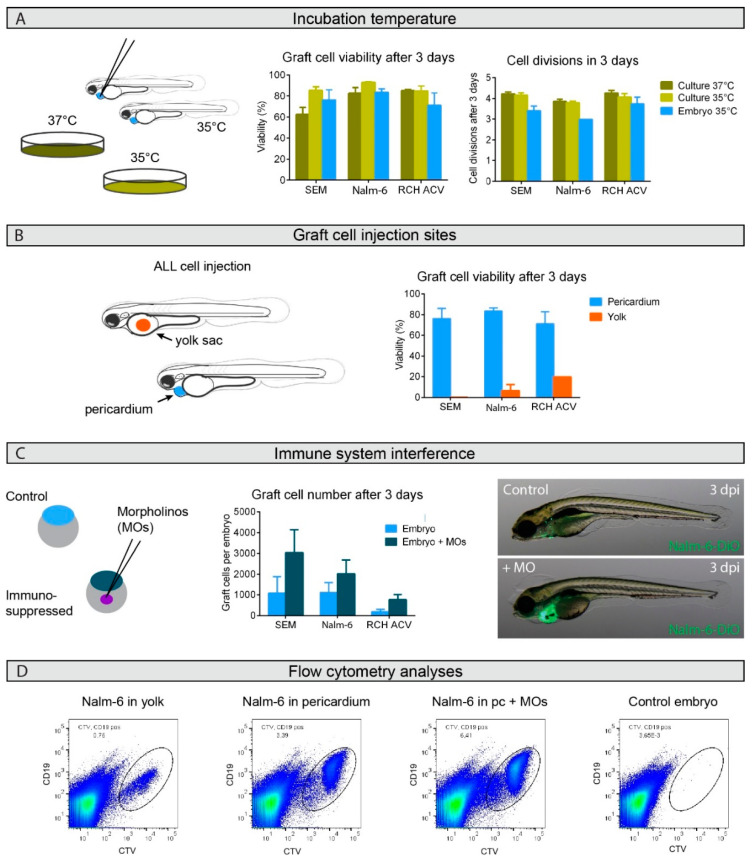
Successful B-cell precursor acute lymphoblastic leukemia (BCP-ALL) graft expansion in zebrafish embryos depends on graft site and host immunosuppression but not on temperature. (**A**). Viability and growth rate were flow cytometrically assessed for graft cells from groups of 10–20 zebrafish embryos at 35 °C or classical in vitro cultures at 35 °C and 37 °C from three or four independent experiments. (**B**). Graft cell viability was flow cytometrically assessed 3 days after injection into either the yolk sac (orange) or pericardium (blue). The two independent experiments performed for injections of Nalm-6 into the yolk sac compared to the four experiments with pericardium injections produced a significant difference, with a *p*-value of 0.007 in an unpaired *t*-test. Injections were performed in groups of 10–20 embryos. (**C**). Host embryos were immunosuppressed by morpholino (MO) injection into fertilized eggs (dark blue) and compared to untreated controls (light blue). Graft cell number per host embryo was calculated from flow cytometric analyses from groups of 10–20 embryos in three or four independent experiments. The comparisons of immunosuppressed injections with untreated controls produced *p*-values in a paired *t*-test: *p* = 0.045 for SEM cells, *p* = 0.095 for Nalm-6 cells and *p* = 0.049 for RCH-ACV cells. Bars represent means ± SEM. Microscopic images show 5-day old host embryos with (+MO) or without (control) immunosuppression 3 days post-injection (dpi) with DiO-labeled Nalm-6 cell injections into the pericardium. Only one biological replicate was performed for SEM and RCH-ACV injections into the yolk sac. Representative images shown. (**D**). Representative flowcytometric scatter plots of Nalm-6 cells following engraftment in zebrafish embryos. CD19 positive Nalm-6 cells prelabeled with CellTrace Violet can be separated from auto-fluorescent zebrafish cells to sort out the graft cell population for analysis. Engraftment site indicated as well as whether the host embryo was transiently immunosuppressed using morpholinos (MOs). Groups of 10 embryos from each treatment group were pooled before single-cell dissociation for flow cytometric analysis. Control embryos not engrafted show auto-fluorescence. For details see also Appendix A. Pc = pericardium.

**Figure 2 cancers-12-01883-f002:**
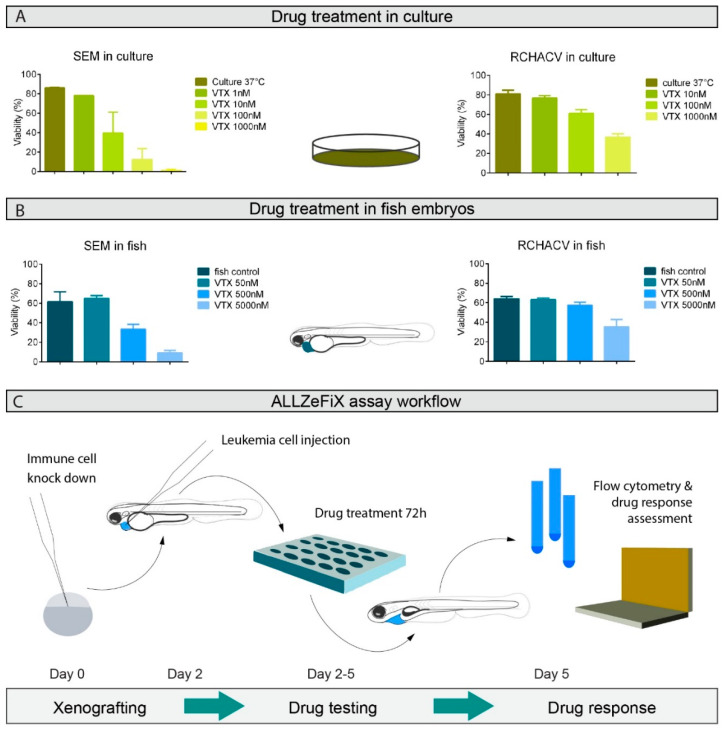
Optimized human–zebrafish xenograft (ALL-ZeFiX) assay reveals graft response to venetoclax (VTX) comparable to results from conventional in vitro treatment. (**A**). Viability of SEM and RCH-ACV cells treated for 48 h with indicated concentrations of venetoclax at 37 °C from two or three independent experiments in conventional 2D culture assessed by flow cytometry. (**B**). Viability of SEM and RCH-ACV cell grafts in immunosuppressed embryos treated with indicated concentrations of venetoclax at 35 °C for 72 h from three independent experiments assessed by flow cytometry. Each experiment was measured as the mean of a pool of 10–20 embryos. Bars represent means ± SEM. (**C**). Workflow is diagrammatically shown for the ALL-ZeFiX assay.

**Figure 3 cancers-12-01883-f003:**
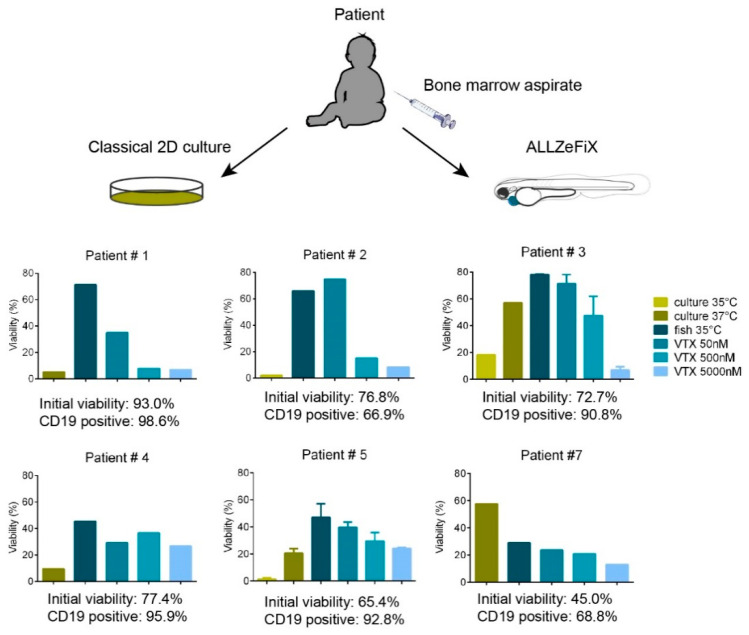
Primary BCP-ALL patient samples can be expanded in zebrafish embryos with individual responsiveness to venetoclax. Viability of isolated mononuclear cells from primary bone marrow aspirates engrafted in immunosuppressed zebrafish embryos or in vitro 2D cultures as controls were assessed flow cytometrically after 3 days of exposure to the range of venetoclax concentrations. Graphs for patients 1, 2, 4 and 7 represent one experiment measured as the mean of a pool of 10–20 embryos. Graphs for patients 3 and 5 represent the means from two independent experiments. Error bars represent SEM.

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
