# Peer review of "Fast, In Vivo Model for Drug-Response Prediction in Patients with B-Cell Precursor Acute Lymphoblastic Leukemia"

_cancers, 2020, doi:10.3390/cancers12071883_

Round 1

Reviewer 1 Report

I believe the manuscript improved greatly and is a very important study.

I would just would like to point that the main figures would gain if they would have some FACS plots- one example i.e not put all in Supplementary- I think it reflects more the rigorous and hard work behind the final graphs and would benefit the paper and the readers.

MINOR:

Is missing the concentration of the cells when injected- at least I could not find it.

Author Response

Answer to Rev.1:

Dear Reviewer # 1,

Thank you very much for your comments on our revised manuscript, which we address below.

  1. Following the reviewer’s suggestions, we added representative scatter blots related to experiments in Figure 1 as Figure 1D in the revised manuscript now.
  2. We apologize for the missing information about concentration of injected cell suspension. We added the following information to the Methods section in line 355, where we state: “…to inject 300-500 human BCP-ALL cells at a concentration of 2*105/ml into each 48 hours post fertilization embryo.”

Reviewer 2 Report

I thank the authors for answering all my comments. I have no more scientific concerns. The authors may only want to be careful with the spelling of genes and transgenic lines. Please refer to https://wiki.zfin.org/display/prot/Conventions+For+Naming+Zebrafish+Genes. 

Additionally, the Nalm-6 cell lines is known as such and not as Nalm6. Please update if possible. 

Author Response

Answer to Rev.2:

Dear Reviewer # 2,

Thank you very much for your comments on our revised manuscript, which we address below.

  1. Following the reviewer’s suggestions, we corrected the spelling of all gene, protein and transgenic line names and in Figure S4 of the revised manuscript.
  2. We also corrected the spelling for the cell line to Nalm-6 in all cases in the text of the revised manuscript and the figures 1 and S1-S4 now.

This manuscript is a resubmission of an earlier submission. The following is a list of the peer review reports and author responses from that submission.

Round 1

Reviewer 1 Report

It was not a reliable in vivo system, and low quality and deficiency of the present data.

Author Response

We did not find any constructive critics or questions in this “review” that could be answered.

Reviewer 2 Report

Fast, in vivo model for drug-response prediction in patients with B-cell precursor acute lymphoblastic leukemia

In this work authors explore the zebrafish xenograft model to screen drugs for B-cell precursor (BCP) acute lymphoblastic leukemia (ALL) patients. They develop a new method, site of injection as well as quantification by FACS. They also inject in MO- based innate immune compromised fish in a 5-day ALL-ZeFiX assay.

In general, very interesting results and data but need further clarification of some points:

MAJOR

  1. Show FLOW CITOMETRY gatings - do not refer to the use of CD19 antibody in the text – need to describe better the strategy in the main text and show the grafts – also what is the % of CD19 in each BM sample and corresponding initial viability- need to show in the main figure. Are you accessing viability of only CD19 cells?or overall?not clear.
  2. Viability – show that is targeting the CD19 populations and not others
  3. No statistical analysis in Fig1C.
  4. Figure 2 was not clear that the fish were morphants. I assume they were …not clear in figure 3 I could find in the legend but not in Fig.2
  5. Figure 2 A/B show dot plots to see dispersion
  6. Cells do not go into circulation?can you explain this? What have you seen?
  7. In line 160/161..... “3 responded poorly (patients 4, 5 and 7 with IC50 ~5000nM, Figure 3)”. But then you contradict your selves saying that Patient 7 responded well:.. ”Engrafted cells from patient 7 responded to venetoclax (IC50 between 500 and 5000nM), in line with a good clinical response of this patient in the ongoing trial”” (Suppl. Figure 5)??? I am sorry but this seems a contradiction. Also, you say is poor responder because only responds at high concentration but then is good responder because the patient responds well?

In patients, maximum plasma concentrations goes 1-30uM (what I was able to see) so maybe your gradient of concentrations might be useful…but also confusing because your Patient7 is responding at 5000nM =5uM- at the range of the patient concentration dose…?

MINOR

- why use 96 well plates to then pull the xenografts – can you explain the rational

- small Introduction – maybe expand a bit more.

- We treated 7 patient-derived xenografts with three concentrations between 50 and 5000nM 157 (Figure 3) or 10 and 1000nM (Suppl. Figure S4A) of venetoclax- Why??

- line 141- no call to figure

-please describe the clinical history of the main patients at least – and when did you had access to the sample – before any treatment? Naïve? Not referred

- Please plot your viability plots normalized to control - in this way we can evaluate much better the fold decrease – or have graphs side by side

- Also, you do not comment that cells in general have a higher viability in the fish in vivo than in vitro

-Discussion- “The ALL-ZeFiX assay improves on in vitro co-cultures”- What do you mean please explain better

-“The hypoxic 192 conditions might explain, why any BCP-ALL cells transplanted into the yolk sac survived”. Typo ? or no?

- Please discuss the need to inject morpholinos – try to use mutants ?? could be a more feasible way.

Author Response

Dear Reviewer # 2,

Thank you very much for your detailed comments on our manuscript, which we address point-by-point below.

The revised manuscript was edited by a native English-speaking scientist.

  1. Six new supplementary figures (S1, S6-S9 and S11) show gating strategies and scatter plots to show results for the grafts, as the reviewer suggested. Figure S1 e.g. presents the gating strategy in experiments examining the expansion of the Nalm6 cell line in culture and in engrafted embryos to accompany results depicted in Figure 1. We have revised the explanation of our gating strategy and how we assess viability and assure that only human graft cells are being analyzed in the detailed legends for the supplementary figures as well as lines 115-125 where we state, "Graft cells were labeled with a fluorescent proliferation marker (CellTrace Violet) prior to transplantation, to support quantification of graft cell proliferation over time. CD19 expression was also assessed to monitor BCP-ALL blast cell identity prior and post engraftments. After 3 days at 35°C, 10-20 engrafted embryos were pooled and dissociated into single-cell suspensions, in which leukemia cell viability, proliferation and total human leukemia cell numbers were flow cytometrically analyzed (Suppl. Figure S1) to assess engraftment success and graft expansion. Viability and CD19 expression were directly measured from the CTV-positive graft cell population (see gating strategy, Suppl. Figure S1). Counting only CD19-positive cells assured that we only assessed engrafted human leukoblast expansion. The number of cell divisions of the CD19-positive leukoblasts was calculated from the fluorescence intensity of the CellTrace violet proliferation marker, which is reduced by half in successive daughter cell generations.” The micrograph in Figure 1C is a representative example of the engrafted fish. Since the engrafted embryos are rapidly dissociated into single-cell suspensions for FACS-based analysis of the light-sensitive CellTrace Violet-labeled graft cells, no microscopic images are included as regular documentation in the experiments.

The range of CD19-positive cells in the bone marrow samples is now included in the revised manuscript (line 204) with the range of initial viability and CD19-positive proportion. We also added exact values for each patient sample in Figure 3 and Supplementary Figure S10 as the reviewer requested.

  1. The new figures S1, S6-S9 and S11 of the revised manuscript with FACS gating strategies now show the detection of CD19 expression, which was assessed prior and following the sorting of the graft cell population in which viability was assessed. We cannot exclude that venetoclax also affects a CD19-negative fraction of mononuclear cells from primary human bone marrow samples.
  2. We describe the statistical analyses for Figure 1C with p-values for graft cell number enhancement due to immunosuppression of the host in the legend for Figure 1.
  3. We apologise for the missing information in Figure 2 of the original manuscript. The embryo avatars used/presented in Figures 2 and 3 are indeed morphants. After determining that morpholino-based transient immunosuppression was necessary for optimal graft survival and growth in the ALL-ZeFiX assay (Figure 1), all further experiments were conducted using morpholino-based transient immunosuppression. To make this clearer to the reader, we include information about whether the embryos are immunosuppressed in every legend for figures showing data from ALL-ZeFiX assays and included the sentence, “Therefore, all further experiments using the ALL-ZeFiX assay were conducted in morpholino-based transiently immunosuppressed zebrafish embryos.” on line 176-177 in the revised manuscript.
  4. Following the reviewer suggestion, we added scatter plots showing dispersion for flow cytometric counts of venetoclax-treated cell culture and xenografts as Supplementary Figures S6-9 for 4 representative experiments included in the results described in Figure 1A and B.
  5. We only microscopically observed graft expansion in the initial experiments to optimize engraftment. All testing using the ALL-ZeFiX assay was designed to assess graft expansion in a purely quantitative manner from the single-cell suspensions from pooled embryos as readout for the effect of drug treatment. From the initial observation, graft cells were dispersed primarily to the rostral part of the non-immunosuppressed embryo, as shown in Figure 1C, with no obvious contribution of the blood circulation. We did not microscopically monitor the precise locations of disseminated cells in host embryos, because this was not necessary to use the ALL-ZeFiX assay as a monitor of graft response to the administered drug.
  6. We thank the reviewer for pointing out this misleading and unprecise formulation. We have corrected the wording now in lines 210-214 for a better description of patient response. We write now in the revised manuscript: “In the ALL-ZeFiX assay, 4 of 7 patient-derived BCP-ALL grafts responded very well to venetoclax (patients 1, 2, 3 and 6 with IC50 in the range of 50-500nM, Figure 3 and Suppl. Figure S10A and S11), 2 responded well (patients 5 and 7 with IC50 ~5000nM, Figure 3) and 1 responded poorly after an initial effect at 50nM (patient 4, range of tested concentrations were not active enough to calculate an IC50, Figure 3).”
  7. Doses were administered to patients 4 and 7 as recommended in Place et al., 2018 [doi:10.2217/fon-2018-0121]. Plasma concentration in patients after orally given tablets, which depend on age, weight and liver function and on pharmacokinetics of the drug, have not been assessed in the clinical departments in our patients. Drug safety and efficacy will be assessed in the respective phase I/II trial, and the optimal doses for children will be determined from pharmacokinetic modeling of the trial data. Concentrations tested in our in vivo fish model cannot be directly transferred (1:1) to patient plasma concentration, and we are not attempting to do this. The assay results are intended to provide only a relative prediction of the potential effectiveness of the drug in the patient.
  8. We used 96-well plates hosting one fish per well to prevent cross-contamination from single embryos. Signaling or harmful molecules could be released from dead or dying embryos to induce chain reactions in surrounding embryos. To prevent this possible effect, embryos are reared alone before pooling.
  9. We expanded the introduction now with more background information about the advantages and state of the art for the use of zebrafish embryos as avatars for human cancer cells, as also requested by the editor.
  10. The bone marrow sample from patient 6 was among the first patient-derived xenografts tested in the ALL-ZeFiX assay, and this testing occurred before we settled on a routine drug concentration range. Even though this was treated with different concentration range, we still wanted to include this result to expand the cohort size for this manuscript.
  11. The line noted by the reviewer (now line 180) is a conclusion at the end of a paragraph, and the sentences preceding it in the paragraph all cite the figures that they should.
  12. We revised the Materials and Methods section lines 309-312 to include the relevant clinical history for patients #4 and 7, who are the only two patients whose course of disease could be compared to sample response to venetoclax in our ALLZeFiX assay. It now reads, "Samples from patients #4 and #7 were collected at diagnosis of second relapse. The second relapse was diagnosed in patient #4 approximately one year after completion of first relapse treatment. The second relapse was diagnosed in patient #7 during ongoing disease progression that followed different courses of chemotherapy to treat the first relapse." Relevant clinical history from these two patients was also added to the revised legend for Suppl. Figure S12. Collection timepoint and other patient sample characteristics are now added to Supplementary Table 1.
  13. We understand the intention/rational of the reviewer asking to normalize viability plots to control, but we prefer to keep absolute values since otherwise this would disguise important information to evaluate absolute viability and compare engraftment success with conventional 2D culture conditions. We think, this is particularly important for the evaluation of primary sample engraftment success. We always ran ALLZeFiX graft viabilities and their conventional 2D culture controls in parallel, and show these results in the same graph. To make comparing results as easy as possible, we showed viability plots for xenografts with a maximum of 100% on the y-axes and plotted results from all primary ALLZeFiX grafts from bone marrow samples with a maximum of 80% viability. We have not changed these figures for the reasons stated above, but would be willing to provide normalized viability plots as a supplementary figure if the editor believes this is necessary.
  14. We thank the reviewer for this comment and added this information. The revised text (line 203-208) now reads, “Blast cell viability was 45-93% with 67-99% CD19-positive cells prior to engraftment, and 30-80% with 57-96% CD19-positive cells following 72h engraftment in untreated controls (Figure 3 and Suppl. Figure S6). Viable blast cells completed 1.5 to 3 divisions in the 3-day assay period (Suppl. Figure S6). Blast cell viability was substantially better in the ALL-ZeFiX assay than in 2D cultures (viability ranged from 1-20%) in 6 of 7 patient-derived samples.”
  15. We have omitted the unclear sentence noted by the reviewer at the beginning of the Discussion, and have completely revised the beginning section to better present our thinking. It now reads, "In vitro assays assessing proliferation to preclinically assess drug efficacy on viability of leukemia cells from bone marrow biopsies have improved over the years [2], and have entered clinical use for severe single cases within trials for relapse/refractory disease [3]. One study has described co-culturing primary BCP-ALL blasts with mesenchymal stem cells to improve in vitro culture methods, but these demonstrated high variability of spontaneous apoptosis after 3 days in culture or strongly reduced proliferation [2]. Our own attempts to apply a method of 2D co-culture with patient derived mesenchymal stem cells have not yet been systematically compared to the ALLZeFiX model, however, we show here that viability of patient-derived BCP-ALL blasts is improved in our ALLZeFiX model in comparison to standard in vitro culture. Viability was so high, and with less variability in patient-derived samples in this small pilot cohort that we speculate it could also improve on co-culture with mesenchymal stem cells, since the erratic apoptosis-inducing signals appear to be absent or at least reduced in the 72h zebrafish model”.
  16. We could not find a typo in the sentence quoted by the reviewer from line 192 in our manuscript.
  17. We agree with the reviewer that using mutants rather than injecting morpholinos would indeed be more feasible and preferable. However, our experiments with single morphants suggest the necessity of double morphants in order to reach sufficient graft cell survival even though the effect of Spi1 knockdown shows a more pronounced effect on graft cell survival than Csf3r knockdown (these results are now included in the revised manuscript Supplementary Figure S2). Double mutants are unfortunately predicted to be early lethal (personal communication with Jean-Pierre Levreaud (Institute Marie Curie, Paris, France). One available mutant that provides this global innate immunosuppression is the “cloche” mutant that affects both the endothelial and hematopoietic lineages at a very early stage and is missing the endocardium, the endothelial lining of the heart and dies unfortunately during early development (personal communication, Salim Abdelilah-Seyfried, University of Potsdam, Berlin, Germany), too early for our 5-day assay. The transient effect of Morpholino knockdown was the next best solution for our purpose. We have revised this part of the discussion (line 246-251) to bring out this background information, "Using a method not previously applied for xenografting cancer cells, we improved BCP-ALL cell engraftment conditions by a morpholino-driven temporal immune suppression that blocked innate differentiation of both macrophage and neutrophils. While the use of a mutant zebrafish line lacking both macrophages and neutrophils would be preferable, none exist to our knowledge since knocking out both Spi1 and Csf3r does not produce viable mutants (personal communication with Jean-Pierre Levraud).”

Reviewer 3 Report

In this manuscript, the authors describe the feasibility of using zebrafish PDX models for the transplantation of BCP ALL cells and drug-response prediction in patients. They found that transplantation into the pericardium was superior to transplantation into the yolk and that transient knockdown of spi1 and csf3r allowed better engraftment of these cells. Finally, they confirmed with both cell lines and primary patient material that their model is suitable for the use of drugs in general and for drug-response prediction in particular. This is an interesting finding and I have only a few comments. 

  • Even though I believe the MOs work correctly from the data the authors present, they should nevertheless demonstrate that their phenotypes are specific for the knockdown of their two genes. Control MOs should be injected side by side instead of using uninfected controls, WISH should be performed in order to show that neutrophils and macrophages are indeed diminished, and depending of the kind of MO the authors use, PCR or Western based approaches used to show splice modification or block of translation. Additionally, it might be interesting to see if upon co-injection of wt spi1 or csf3r, mRNA grafts were again as compared to the control situation.
  • the type of MO should be indicated and sequences mentioned. 
  • Additionally, since the authors mention that macrophages are more attracted to the graft side, did the authors consider to specifically knockdown either spi1 or csf3r alone to investigate if both are indeed necessary at the same time or if spi1 knockdown alone might be already sufficient? 
  • For future experiments, do the authors plan using knockout fish instead of using MO-based approaches for each of their screenings? There might e.g. spi1 knockout fish be available from the Sanger zebrafish mutant sequencing project that could be tested for such approaches as well. Alternatively, the authors might want to consider using CRISPR-Cas9 based approaches. This might be something to discuss, since MO injection is not always reliable to the same extent from experiment to experiment.
  • What exactly was analyzed in Figure S3? Did the authors only test for survival of the fish or were any other drug related effects in (the development of) the fish investigated?
  • Regarding their flow cytometric analyses, could the authors provide gating strategies and show representative FACS blots for their final analysis of human grafts into zebrafish? This might be of interest to a lot of zebrafish researchers using the zebrafish as a model for xenotransplanting human cells, since many claim that it is not possible to detect human cells via flow cytometry in fish. 

Author Response

Dear Reviewer # 3,

Thank you very much for your detailed comments on our manuscript, which we address point-by-point below.

We expanded the introduction now with more background information about the advantages and state of the art for the use of zebrafish embryos as avatars for human cancer cells, as also requested by the editor.

  1. We apologize for the confusion about missing control experiments. The mistake was here on our side due to incomplete citation that we used by mistake to refer to the previously established Spi1/Csf3r morpholino function and usage. We have corrected this now by adding two further citations (11 and 13) in the revised manuscript. Spi1 and csf3r (pu.1) gene function and specificity in zebrafish were characterized first in Ellet et al. (ref. 12) and Rhodes et al. (ref. 13) and the morpholinos were described first by Pase et al. in 2012 (ref. 11). All controls mentioned by the reviewer or their equivalent, were conducted in these initial descriptions of the use of these morpholinos. We confirmed mopholino knockdown efficiency in our macrophage (tg(mpeg1-mCherry)) and neutrophil (tg(mpx-EGFP)) reporter lines, which did not show any fluorescent immune cells until day 4 (4dpf) of development after morpholino injection. We included visual evidence for the absence of macrophages in Supplementary Figure S4. After verifying that the morpholinos worked as previously described in our hands, we refrained from using control morpholinos in parallel injections to reduce experimental size.
  2. Morpholino type and sequences were published previously by Pase et al., 2012. We added this information to the Materials & Methods section of the revised manuscript.
  3. As the reviewer requested, we now include a data set in Supplementary Figure S2 showing engraftment results for two cell lines in transiently immunosuppressed embryos that only lack either macrophages (Spi1 Morpholino) or neutrophils (Csf3r Morpholino). We indeed observed a positive effect in Spi1 single morphants on graft cell survival, as the reviewer suspected and as predicted from our microscopic observations. Nevertheless, double morphants clearly enhance graft survival and were therefore chosen for our standard protocol.
  4. We agree with the reviewer that using mutants rather than injecting morpholinos would indeed be more feasible and preferable. However, our experiments with single morphants suggest the necessity of double morphants in order to reach sufficient graft cell survival even though the effect of Spi1 knockdown shows a more pronounced effect on graft cell survival than Csf3r knockdown (these results are now included in the revised manuscript, Supplementary Figure S2). Double mutants are unfortunately predicted to be early lethal (personal communication with Jean-Pierre Levreaud (Institute Marie Curie, Paris, France), as would also a CRISPR knockout. One available mutant that provides this global innate immunosuppression is the “cloche” mutant that affects both the endothelial and hematopoietic lineages at a very early stage and is missing the endocardium, the endothelial lining of the heart and dies unfortunately during early development (personal communication, Salim Abdelilah-Seyfried, University of Potsdam, Berlin, Germany), too early for our 5-day assay. The transient effect of morpholino knockdown was the next best solution for our purpose. We have revised the discussion (line 246-251) to include this background information, "Using a method not previously applied for xenografting cancer cells, we improved BCP-ALL cell engraftment conditions by a morpholino-driven temporal immune suppression that blocked innate differentiation of both macrophage and neutrophils. While the use of a mutant zebrafish line lacking both macrophages and neutrophils would be preferable, none exist to our knowledge since knocking out both Spi1 and Csf3r does not produce viable mutants (personal communication with Jean-Pierre Levraud).”
  5. To evaluate host toxicity to the drugs used, our experiment was devised to test lethality as the only criterion for Figure S3.
  6. We now include 6 additional figures in the revised supplementary material (S1, S6-S9 and S11) that show the FACS analyses scatter plots. Figure S1 presents the gating strategy in experiments examining the expansion of the Nalm6 cell line in culture and in engrafted embryos to accompany results depicted in Figure 1. Figure S11 demonstrates our FACS results for the venetoclax-treated bone marrow sample from patient 3 to accompany results presented in Figure 3. All parameters we describe in the text are assessed only from CellTrace Violet fluorescently labeled graft cell populations (labeled prior to engraftment), with a final subsequent evaluation of CD19 identity prior to FACS analyses (in the single-cell suspension resulting from the ALL-ZeFiX assay). We have revised the explanation of our gating strategy and how we assess viability and assure that only human graft cells are being analyzed in lines 115-125 where we state, "Graft cells were labeled with a fluorescent proliferation marker (CellTrace Violet) prior to transplantation, to support quantification of graft cell proliferation over time. CD19 expression was also assessed to monitor BCP-ALL blast cell identity prior and post engraftments. After 3 days at 35°C, 10-20 engrafted embryos were pooled and dissociated into single-cell suspensions, in which leukemia cell viability, proliferation and total human leukemia cell numbers were flow cytometrically analyzed (Suppl. Figure S1) to assess engraftment success and graft expansion. Viability and CD19 expression were directly measured from the CTV-positive graft cell population (see gating strategy, Suppl. Figure S1). Counting only CD19-positive cells assured that we only assessed engrafted human leukoblast expansion. The number of cell divisions of the CD19-positive leukoblasts was calculated from the mean fluorescence intensity of the CellTrace violet proliferation marker, which is reduced by half in successive daughter cell generations.”